# PTX-3 Secreted by Intra-Articular-Injected SMUP-Cells Reduces Pain in an Osteoarthritis Rat Model

**DOI:** 10.3390/cells10092420

**Published:** 2021-09-14

**Authors:** Minju Lee, Gee-Hye Kim, Miyeon Kim, Ji Min Seo, Yu Mi Kim, Mi Ra Seon, Soyoun Um, Soo Jin Choi, Wonil Oh, Bo Ram Song, Hye Jin Jin

**Affiliations:** Biomedical Research Institute, MEDIPOST Co., Ltd., Seongnam-si 13494, Korea; lmj262@medi-post.co.kr (M.L.); haha38@immunique.co.kr (G.-H.K.); eldjfls3@medi-post.co.kr (M.K.); peacemaker@medi-post.co.kr (J.M.S.); ymkim@medi-post.co.kr (Y.M.K.); seon0735@medi-post.co.kr (M.R.S.); ssoso23@medi-post.co.kr (S.U.); sjchoi@medi-post.co.kr (S.J.C.); wioh@medi-post.co.kr (W.O.)

**Keywords:** SMUP-Cells, mesenchymal stem cells, cell therapy, PTX-3, osteoarthritis, pain relief, macrophage polarization, bioreactor

## Abstract

Mesenchymal stem cells (MSCs) are accessible, abundantly available, and capable of regenerating; they have the potential to be developed as therapeutic agents for diseases. However, concerns remain in their further application. In this study, we developed a SMall cell+Ultra Potent+Scale UP cell (SMUP-Cell) platform to improve whole-cell processing, including manufacturing bioreactors and xeno-free solutions for commercialization. To confirm the superiority of SMUP-Cell improvements, we demonstrated that a molecule secreted by SMUP-Cells is capable of polarizing inflammatory macrophages (M1) into their anti-inflammatory phenotype (M2) at the site of injury in a pain-associated osteoarthritis (OA) model. Lipopolysaccharide-stimulated macrophages co-cultured with SMUP-Cells expressed low levels of M1-phenotype markers (CD11b, tumor necrosis factor-α, interleukin-1α, and interleukin-6), but high levels of M2 markers (CD163 and arginase-1). To identify the paracrine action underlying the anti-inflammatory effect of SMUP-Cells, we employed a cytokine array and detected increased levels of pentraxin-related protein-3 (PTX-3). Additionally, *PTX-3* mRNA silencing was applied to confirm PTX-3 function. *PTX-3* silencing in SMUP-Cells significantly decreased their therapeutic effects against monosodium iodoacetate (MIA)-induced OA. Thus, PTX-3 expression in injected SMUP-Cells, applied as a therapeutic strategy, reduced pain in an OA model.

## 1. Introduction

Mesenchymal stem cells (MSCs) are capable of differentiating into various cell lineages, and their role in regenerative medicine for repairing damaged tissues is well documented. Recent studies have demonstrated that the therapeutic effects of MSCs are mediated by their paracrine activity and that MSCs behave similarly to drugs with complex functions in the microenvironment of various diseases. This suggests the possibility of developing a treatment strategy with MSCs for refractory diseases [1,2]. Previously, intra-articular administration of allogeneic MSCs in a pain-induced monoiodoacetate (MIA)-injected rat arthritis model was found to increase the expression of tumor necrosis factor (TNF)-stimulated gene 6, which has an anti-inflammatory action on cartilage, and decrease the matrix-degrading enzyme disintegrin and metalloproteinase with thrombospondin motifs 5 (ADAMTS5) [3]. This finding suggests that MSCs are capable of not only acting similar to pharmacological drugs at lesion sites to relieve inflammation, suppress pain, and participate in immune responses but also inhibiting degenerative molecule expression involved in OA pathogenesis [1,2,3].

The commercialization and globalization of existing stem cell therapies are restricted by limitations in production technology and formulation [4]. Additionally, the aging of stem cells in simple in vitro culture affects the production scale and decreases the efficacy of cells [5]. Thus, improvements in fundamental cell-culture technologies are required. This suggests that improvements in downstream processes, which are relatively less developed than upstream culture technologies, are necessary for the development of next-generation stem cell therapy [6,7]. Previously, we selected small-sized cells at the beginning of culture and observed an increased efficacy of stem cells through improved culture techniques, including the use of hypoxic culture and calcium treatment; we also demonstrate the superiority of MSCs in anti-inflammation and immunomodulation [8,9,10,11]. Specifically, we cultured and mass-produced cells in a bioreactor for clinical application to update the entire culture process up to the application of xeno-free cryoprotectants, resulting in increased proliferation and delayed aging of the cells, and improved expression of stem cell function-related genes—*Oct4*, *Nanog*, *Stella*, *Sal4*, and *BMI-1* [9].

Knee OA (KOA) is caused by worn-out joint cartilage or degenerative changes due to local inflammation and pain [12,13]. The primary goal of KOA treatment is to alleviate the pain and stiffness, maintain and improve joint motor skills, and minimize disability. Furthermore, the quality of life of patients needs to be improved by regenerating damaged cartilage and suppressing additional joint damage [13]. KOA is also referred to as degenerative arthritis and is caused by an imbalance between degeneration and regeneration of joint cartilage and bone with impaired intrinsic recovery function [14,15]. In particular, degenerative chondrocytes secrete inflammatory cytokines and increase the expression of various matrix metalloproteinases (MMPs), activated variants of which directly degrade the extracellular matrix of chondrocytes and induce gradual degeneration of cartilage tissues [16,17].

Therefore, in this study, we assessed the potential of using a SMall cell+Ultra Potent+Scale UP cell (SMUP-Cell) platform as a therapeutic agent, applied to control and reduce pain in a rat model of OA. To this end, we established a next-generation stem cell-therapeutic platform that ultimately identified PTX-3 secreted by SMUP-Cells as a marker to screen highly potent cells with excellent therapeutic efficacy for use in inflammatory and pain environments. The anti-inflammatory effects of SMUP-Cells in vivo suggest the efficacy of this method, thereby promoting the application of MSCs in clinical treatment.

## 2. Materials and Methods

### 2.1. SMUP-Cell Platform

This study was approved by the Institutional Review Board of MEDIPOST Co., Ltd. (MP-2015-6-4; MEDIPOST, Seongnam City, Korea). Neonatal umbilical cord blood (UCB) was collected from the umbilical vein following informed maternal consent. Mononuclear cells (MNCs) were isolated from harvested UCB samples by centrifugation using Ficoll-Paque PLUS (GE Healthcare, Uppsala, Sweden). The separated MNCs were seeded at a density of 5 × 10^5^ cells/cm^2^ and maintained under hypoxic conditions (3% O_2_) in minimum essential medium-α (Gibco, Grand Island, NY, USA) supplemented with 1.8 mM calcium. MSC colonies were suspended and counted after the formation of fibroblast-like adherent homo cells. To confirm the changes in differentiation potential by hypoxia conditions, we analyzed for osteogenic, chondrogenic, and adipogenic differentiation, by staining for alkaline phosphatase (ALP), safranin O, and oil red O and found no differences between the two groups (Appendix A). For SMUP-Cells, MSCs were isolated using plurStrainer of pore size 10 µm (pluriSelect, San Diego, CA, USA) to remove large cells [9]. After size filtering, we checked that the majority of small cells (93 ± 1.9%) had a diameter of ≤10um (Appendix A). These cells were counted, sub-cultured at a density of 500~3000 cells/cm^2^ and maintained until passage 5 under the same conditions. For mass production, the cells at passage 5 were transferred into a bioreactor (Pall Life Science, Brussels, Belgium) and cultured until passage 6. The primary validation of the bioreactor control system was to maintain defined environmental factors (pH, dissolved oxygen (DO) level, and temperature) according to the instrument protocol [18] (Appendix A). The cells grown in the bioreactor were monitored by microscopy and then harvested; viable cells were counted by Trypan Blue exclusion. Finally, we created stocks of a xeno-free solution (Sol1, MEDIPOST Co., Ltd., 1 × 10^7^ cells/2 mL) for aseptic applications using a Crystal filling line (Aseptic Technology, Raleigh, NC, USA). SMUP-Cell processing is summarized in Figure 1. Each SMUP-Cell from five donors was used in this study (n = 5) (Appendix A).

### 2.2. SMUP-Cell Characterization

Senescence-associated beta-galactosidase (SA β-gal) staining was used to determine senescence phenotype, with a histochemical staining kit (Cell Signaling Technology, Danvers, MA, USA) employed to confirm senescence-associated beta-galactosidase (SA β-gal) activity according to the manufacturer’s instructions, followed by an examination of the cells using a microscope; 5 fields/cell were assessed [19]. To demonstrate cell-surface antigen expression, cells were stained for 15 min at room temperature using fluorescein isothiocyanate (FITC)-conjugated antibodies against human CD14 and CD45 (BD Biosciences, Franklin Lakes, NJ, USA) and phycoerythrin (PE)-conjugated antibodies against human CD73, CD90, CD166 (BD Biosciences), and CD105 (Invitrogen, Carlsbad, CA, USA). Isotype-matched mouse antibodies were used as controls. All surface markers were diluted in phosphate-buffered saline (PBS, Grand Island, NY, USA) (1 × 10^5^ cells in 10 µL of the sample). The cells were washed with PBS and fixed with 1% (v/v) paraformaldehyde (Sigma-Aldrich, St. Louis, MO, USA). SMUP-Cells were evaluated by flow cytometry and using a FACSCalibur instrument (BD Biosciences), and the percentage of cells expressing surface proteins was analyzed for 10,000 gated-cell events [19]. To assess differentiation ability, SMUP-Cells were stimulated in induction media to induce their differentiation into osteocytes, chondrocytes, and adipocytes [9]. After stimulation, the multi-lineage ability was assessed. Specifically, osteocytes were evaluated by measuring the expression of alkaline phosphatase (ALP) and intensity of von kossa staining (Sigma-Aldrich, Sigma-Aldrich, St. Louis, MO, USA), chondrocytes were confirmed by Safranin O and Collagen type II staining (Sigma-Aldrich, St. Louis, MO, USA), and adipocytes were verified by staining of the accumulated lipid vacuoles with Oil Red O (Sigma-Aldrich, St. Louis, MO, USA) [10]. In addition, we used the fluorescent neutral lipid dye 4.4-difluoro-1,3,5,7,8-pentametyl-4-bora-3a, 4a-diaza-s-indacene (BODIPY 493/503; Molecular Probes, Carlsbad, CA, USA) to confirm lipid droplet formation [20]. Nuclear counter staining was performed with 6-diamidino-2-phenylindole (DAPI, Invitrogen, CA, USA) for 5 min at room temperature in the dark, and fluorescent images were acquired using an LSM 800 confocal microscope (Zeiss, Oberkochen, Germany). We have added the induction medium composition in Appendix A. The senescence of cells was tested by SA β-gal staining (Sigma-Aldrich, St. Louis, MO, USA), following the manufacturer’s instructions.

### 2.3. Inflammation Status under the Co-Culture System

The mouse macrophage cell line RAW 264.7 was obtained from American Type Culture Collection (ATCC, Manassas, VA, USA). The cells were cultured at a density of 1 × 10^5^ cells/well in serum-free Roswell Park Memorial Institute Medium 1640 and stimulated with 1 µg/mL lipopolysaccharide (LPS; derived from Escherichia coli O55:B5; L6529; Sigma-Aldrich, St. Louis, MO, USA). SMUP-Cells (1 × 10^5^) were directly seeded with the LPS-activated RAW 264.7 cells; the supernatants were collected after 48 h and the concentrations of mouse TNFα, mouse arginase-1 (ARG-1), and human pentraxin-related protein-3 (PTX-3) were evaluated using enzyme-linked immunosorbent assay (ELISA; R&D Systems, Minneapolis, MN, USA).

### 2.4. Immunofluorescence Staining

RAW 264.7 cells were diluted with blocking solution (Invitrogen), and then incubated at 4 °C overnight with antibodies against mouse CD11b (Abcam, Cambridge, UK) and mouse CD206 (R&D Systems). After washing, the cells were incubated with Alexa Fluor 488- or cyanine 3 (Cy3)-conjugated secondary antibodies (Jackson ImmunoResearch Europe Ltd., Newmarket, UK) for 30 min at room temperature in the dark. Nuclear counter staining was performed with Hoechst 33342 for 5 min at room temperature in the dark, and fluorescent images were acquired using an LSM 800 confocal microscope (Zeiss, Oberkochen, Germany).

### 2.5. Cytokine Array and Western Blotting

We collected conditioned media from two groups after 48 h of incubation (SMUP-Cell alone or SMUP-Cell with LPS-treated Raw 264.7, n = 1). A human proteome profile array (55-spot; R&D Systems) was used to analyze proteins secreted by SMUP-Cells under inflammatory conditions according to the manufacturer’s instructions (Appendix A). Antibodies were imaged using a ChemiDoc XRS camera (Bio-Rad Laboratories, Hercules, CA, USA). For each spot on the membrane, the optical density was measured using Image Lab software (v6.0.1; Bio-Rad Laboratories) and calculated by normalization to the reference spot. Cell extracts were obtained in a buffer containing 9.8 M urea, 4% 3-[(3-cholamidopropyl)dimethylammonio]-1-propanesulfonate (CHAPS), 130 mM dithiothreitol, 40 mM Tris-HCl, and 0.1% sodium dodecyl sulfate. Protein concentrations were determined using a bicinchoninic acid assay kit (Sigma-Aldrich). Protein extracts (15 μg) were separated by sodium dodecyl sulfate-polyacrylamide gel electrophoresis, and the resolved proteins were transferred onto nitrocellulose membranes. The membranes were stimulated with antibodies against p16 and β-actin (Sigma-Aldrich).

### 2.6. Quantitative Real-Time Polymerase Chain Reaction (qRT-PCR) and Small-Interfering (si)RNA Assay

The total RNA was extracted from activated RAW 264.7 cells cultured with SMUP-Cells using TRIzol reagent (Invitrogen), and cDNA was prepared using a cDNA synthesis kit (Roche, Basel, Switzerland). Transcript levels were quantified by qRT-PCR using a LightCycler 480 System (Roche). SMUP-Cells were transfected with 25 nM PTX-3 siRNA and scrambled control siRNA with DharmaFECT reagent (Dharmacon Inc., Lafayette, CO, USA) for 24 h. Naïve cells were cultured in a basic medium without transfection. The primers used for qRT-PCR and the siRNA targeting sequences are provided in Appendix A.

### 2.7. Animals and Induction of OA

All animal experiments in this study were performed according to the guidelines of the Animal Care and Use Committee of MEDIPOST Co., Ltd. (MP-LAR-2017-2-2). Male Sprague–Dawley rats (Samtako Bio Korea Co., Ltd., Osan, Korea) aged 6 weeks (weight: 170–200 g) were used in these experiments. Rats were randomly divided into six groups and treated as described in Appendix A. For monosodium iodoacetate (MIA)-induced OA, rats were anesthetized with 3% isoflurane in O_2_, and OA was induced by intra-articular (IA) injection of 2 mg MIA (I2512; Sigma-Aldrich) dissolved in saline through the infrapatellar ligament of the right knee. SMUP-Cells (2.5 × 10^5^ cells/25 µL) were administered along with 1% sodium hyaluronic acid (HA; Humedix Co., Ltd., Anyang, Korea) pre-treatment by IA route 4 days after the MIA injection (Appendix A). For in vivo imaging, SMUP-Cells treated with Neostatin 749 reagent (NEO Science, Gyeonggi, Korea) at room temperature for 20 min were used. Images were taken using the Cy7 filter of the FOBI instrument (FOBI Ver3.1; NEO Science, Gyeonggi, Korea) and the fluorescence data were analyzed using NEOimage for FOBI software (NEO Science).

### 2.8. Pain Assessment: Measurement of Hind-limb Weight Distribution

Hind-limb weight distribution was measured using an incapacitance meter (Harvard Apparatus, Boston, MA, USA) as an index of pain [21]. Rats were positioned on the incapacitance meter with each hind limb resting on two separate sensor plates. The force exerted by each hind limb was measured five times in grams and averaged over a 10-s period by an observer blinded to the given treatment. Each reading was used to calculate the weight on the ipsilateral limb as a percentage of the total weight distributed by both hind limbs; five obtained values were averaged. The rats were measured before induction of OA and at 1 and 4 days after inducing OA (before cell injection). Additional measurements were performed at 3, 10, 17, and 24 days after cell injection.

### 2.9. Statistical Analysis

Values are presented as mean ± standard deviation (SD) or standard error of the mean (SEM). Statistical analysis was performed using the one-way analysis of variance with Tukey’s least-significant difference post-hoc tests using GraphPad Prism software (v5.0; GraphPad Software, San Diego, CA, USA). For all tests, results with *p* < 0.05 were considered statistically significant.

## 3. Results

### 3.1. SMUP-Cells Possess Stem Cell Characteristics

We previously reported the culture conditions for primitive MSCs [8,9,10,11]. Specifically, small MSCs primed with 3% hypoxia and calcium were added to the cell culture up to passage 6, which improved in vitro stem cell capacity and immunoregulatory effectiveness, as well as their therapeutic function in graft versus host disease mouse model against organ injury [9]. For clinical development, we updated workflow processes, such as bioreactors and frozen preservative solutions (xeno-free cryopreservation), and instructions were included for the automated frozen storage of bulk cultured cells. In the present study, we developed a new protocol (SMUP-Cell) that combines advanced small cell (SM) selection with proprietary culture methods to retain and enhance optimal stem cell characteristics throughout the repeated subculture platform, thereby enabling the manufacture of ultra-potent (UP) cells by scale-up (UP) culture expansion (Figure 1). To demonstrate the SMUP-Cell characteristics, we assessed stem cell properties, such as morphology, senescence phenotype, differentiation potential, and surface antigens. We observed a similar adherent fibroblast-like morphology (Figure 2A). Evaluation of senescence phenotypes by SA β-gal staining revealed no positive staining (Figure 2B). Upon long-term culturing, at passage 16, SMUP-Cells exhibited an enlarged morphology and strong SA β-gal staining. Moreover, the senescence-related protein p16 was not expressed at passage 6. The level of p16 was considerably upregulated during passages 6 to passage 16 (Appendix A). To confirm their multi-lineage potential, SMUP-Cells were incubated in an induction medium and examined by staining with ALP or Von kossa (osteocyte), Oil Red O or BODIPY (adipocyte), and Safranin O or Collagen type II (col II, chondrocyte) as specific lineage markers (Figure 2C). Immunophenotype results showed that SMUP-Cells were positive for the expression of CD73, CD90, CD105, and CD166 but negative for CD14, CD34, and HLA-DR according to standards of the International Society of Cell Therapy (ISCT, Figure 2D) [22]. Collectively, SMUP-Cells exhibited stem cell-like features and potential for use as a therapeutic source.

### 3.2. SMUP-Cells Activate Macrophage Polarization

To verify macrophage polarization activated by SMUP-Cells, we used Raw 264.7 cells (mouse macrophages) exposed to LPS and co-cultured with SMUP-Cells for 2 days. Immunofluorescence staining was used to confirm the presence of M1 inflammatory markers (CD11b, green) or M2 anti-inflammatory markers (CD206, red), and the positive cells were counted. LPS-exposed Raw 264.7 cells displayed considerable upregulation of CD11b; however, LPS-stimulated Raw264.7 cells co-cultured with SMUP-Cells revealed blockage of CD11b expression and significant upregulation of CD206 expression relative to the levels in control Raw 264.7 cells (Figure 3A). We then analyzed gene-expression levels of pro-inflammatory (mouse IL-1b, IL-6, and TNF-α) and anti-inflammatory (mouse ARG1) cytokines in Raw 264.7 using qRT-PCR. The results indicated induced expression of proinflammatory cytokines following LPS treatment, although this induction was significantly blocked by co-culture with SMUP-Cells, which promoted significantly upregulated expression of anti-inflammatory cytokines relative to those in the control Raw 264.7 cells (Figure 3B). To confirm the levels of secreted cytokines (TNF-α and ARG-1) in each culture supernatant, ELISA was performed. The results revealed increased levels of TNF-α following LPS induction, whereas co-culture with SMUP-Cells significantly inhibited these levels and promoted ARG-1 secretion from macrophages (Figure 3C). These results demonstrated that SMUP-Cells promoted macrophage polarization to an anti-inflammatory phenotype under inflammatory conditions in vitro.

### 3.3. PTX-3 Secreted by SMUP-Cells Is Critical to Macrophage Polarization

Various soluble factors reportedly control the beneficial functions of MSCs, including tissue regeneration [23]. Here, we identified an array of factors secreted by SMUP-Cells as candidate markers for macrophage activation in cultured media from each experimental group (SMUP-Cells only and SMUP-Cell/Raw 264.7+LPS co-culture, Figure 4A). We identified eight secretory proteins considerably upregulated in SMUP-Cells by co-culture—granulocyte-macrophage colony-stimulating factor (GM-CSF), tissue inhibitor of metalloproteinase-1 (TIMP-1), PTX-3, insulin-like growth factor-binding protein (IGFBP)-1, thrombospondin-1 (TSP-1), IGFBP-3, fibroblast growth factor-7, and angiopoietin-1 (Ang-1) (Figure 4B). Among these, we specifically analyzed the increase in the secretion of anti-inflammatory paracrine factors (PTX-3, TIMP-1, TSP-1, and Ang-1) by an array spot. Importantly, PTX-3 showed the greatest upregulation under inflammatory conditions. To evaluate PTX-3 secretion, we measured the protein levels under each of the four lot conditions by ELISA. PTX-3 secretion was not detected in Raw 264.7 cells or those exposed to LPS; however, PTX-3 secretion was significantly promoted in co-cultures of SMUP-Cells and macrophages, at least 9.4-fold higher than that observed in cultures of SMUP-Cells only (Figure 4C). Based on these findings, we selected PTX-3 for further investigation of its role in inflammation-mediated SMUP-Cell activity. 

To confirm that PTX-3 functionally contributed to macrophage polarization initiated by SMUP-cells, we inhibited PTX-3 expression using siRNA. Control data showed that treatment with the target siRNA significantly blocked PTX-3 expression at the gene and protein levels, as shown by qRT-PCR and ELISA (Figure 5A,B).

To determine the changes in SMUP-Cell features by silencing PTX-3, we analyzed the morphology and MSC-specific markers and found no differences between the experimental groups (Appendix A). After a 48-h co-culture to assess macrophage polarization, Raw 264.7 cells were stained for the M1 pro-inflammatory marker CD11b or the M2 anti-inflammatory marker CD206, and the positive cells were counted. The results showed that PTX3 silencing in SMUP-Cells significantly upregulated CD11b expression and reduced CD206 levels relative to those in the controls (naive SMUP-Cells or SMUP-Cells transfected with control siRNA, Figure 5C). Moreover, we found that PTX-3 silencing increased. 

### 3.4. PTX-3 Improves the Therapeutic Capacity of SMUP-Cells in an OA Rat Model

A previous study reported that the injection of increasing concentrations of MIA into joints causes persistent inflammation, which induces structural changes and persistent joint pain [21,24]. Here, we investigated the ability of PTX-3 to control pain relief in an OA rat model. First, to induce an OA model, 2 mg MIA was injected into the right knee of Sprague–Dawley rats, and at 4-days post-injection, non-invasive in vivo detection was carried out by IA injection of Neostatin 749 dye-labeled SMUP-Cells into the knee joint of the rats. Fluorescence signals of SMUP-Cells in the knee were determined at week 0 through week 7 after cell IA injection (Appendix A). Upon injection of labeled SMUP-Cells with or without HA, they remained relatively stable in the knee joints after 4 weeks and became undetectable after 7 weeks (Appendix A). In particular, we confirmed that the integrated density of SMUP-Cells immediately after injection without HA was lower than that using HA. This suggests that, considering the period in which the transplanted cells in the joint were observed, the in vivo efficacy of SMUP-Cells could be maintained for at least 4 weeks. 

We then investigated the significance of PTX-3 in enabling the control of pain relief in an OA rat model. We prepared different types of SMUP-Cells (naïve, transfected with control siRNA, and those transfected with PTX-3 siRNA) and injected them into a rat model of severe OA, and then compared the therapeutic effects. To analyze the therapeutic effect of PTX-3 on MIA-induced joint pain, cells from all groups were respectively transplanted into the knee joints at 4 days after MIA injection along with HA. Pain-related behavior (i.e., asymmetric weight distribution) observed starting from 1 day after MIA injection and continuing for 28 days revealed that MIA stimulated weight-bearing distribution in the rats, which was generally significantly increased in the control SMUP-Cell groups (naïve or control siRNA), but not in the group administered PTX-3 siRNA. The PTX-3 siRNA SMUP-Cell injected OA rats demonstrated similar results to the OA control group without SMUP-Cell treatment. Importantly, the significant increase in weight-bearing distribution in rats administered control SMUP-Cells (naïve and control siRNA) after cell injection subsequently returned to the normal control levels within 14 days (Figure 6A,B). In detail, PTX-3 siRNA SMUP-Cell injected OA rat had a lower level of pain assessment than control SMUP-Cell groups during 14d to 28d (Figure 6B). Basic data related to the pain-related behavior is described in Appendix A. These data demonstrated that PTX-3 secretion played a critical role in relieving arthritis pain. Taken together, our findings suggest that inhibiting PTX-3 expression accelerated the therapeutic effect of SMUP-Cells in OA treatment.

## 4. Discussion

A standardized protocol is required to develop MSC therapeutic products, and it must be applied to Good Manufacturing Practices (GMP). Commercialization of previously developed MSCs is limited due to their monolayer culture, manufacturing processes, high manufacturing costs, and quality control. Therefore, it is necessary to develop next-generation treatments through life-cycle improvement and cost reduction by enhancing cell efficacy beyond the level of simple cell culture. In this study, we isolated small-sized cells and established conditions for the stable culture of cells with stem cell properties under a hypoxic and calcium-rich environment and using a large-scale production method by culturing in a bioreactor. Furthermore, xeno-free preservatives were used to help develop cryo-formulated cell-therapy products. In the early stages of the development of therapeutic agents, studies focused on the process of selecting excellent cells upstream of life-cycle processes. Recently, the interest in downstream product development has increased, leading to various products that require commercialization by researchers, as well as industries. Additionally, the skill of technicians is important in developing effective MSC therapies; however, it is also essential to automate equipment and increase the size of production in order to ensure product uniformity during mass production. Although there are difficulties in developing bioreactors for the mass production of MSCs due to their characteristics as adherent cells, related products have been developed by several companies; furthermore, studies focused on the application of bioreactors for stem cell culture have been published [25,26,27]. These systems can reduce production costs by increasing the batch size and quality of the cell-culture process through automated systems that enable mass production [28]. The bioreactor used in the present study was capable of real-time monitoring of culture conditions (DO, pH, and temperature) [18,29,30,31]. Recently, numerous serum-free, xeno-free, and dimethyl sulfoxide-based cryopreservation solutions have been developed; however, the recovery rate of cells from these solutions is low, suggesting that improvements are needed. In the present study, we developed a dispensing system to automate the process of freezing mass-produced cells and applied it to the SMUP-Cell culture. It improved the dispensing efficiency in terms of accuracy, reproducibility, time, and the required number of technicians compared with conventional manual work. To develop SMUP-Cell therapeutics, we are in the process of constantly updating the above process.

Arthritis begins with inflammation of the synovial membrane, which causes structural damage to the joint and pain, thereby interfering with joint movement [32]. Here, we observed that SMUP-Cell reduced inflammation in vitro. The application of a protein array to evaluate proteins secreted by SMUP-Cells and possibly responsible for macrophage polarization in the inflammatory environment revealed multiple upregulated proteins in the co-culture medium relative to that in the culture medium of SMUP-Cells alone. Among the four-candidate anti-inflammatory paracrine factors, PTX-3 was the most highly and consistently upregulated under all experimental co-culture conditions. TIMP-1, TSP-1, and Ang-1 are secreted from UCB-derived MSCs and participate in MSC-related therapeutic activity [33,34,35]; therefore, these findings suggest that additional potency markers likely exist.

PTX-3 induces the polarization of macrophages to an M2 anti-inflammatory phenotype along with the secretion of immunomodulatory cytokines [36]. MSCs trigger changes in macrophage polarization toward a proinflammatory M1 or anti-inflammatory M2 phenotype under inflammatory conditions via cell-to-cell contact, paracrine effects, and a combination of these actions [37]. M1 macrophages promote local inflammation by secreting pro-inflammatory cytokines IL-1, IL-6, IL-8, and TNF-*α*, whereas M2 macrophages release anti-inflammatory cytokines (IL-10 or ARG-1), which enable tissue repair in an injured environment [38]. Anti-inflammatory factors secreted by MSCs are responsible for alterations in the immune response [23]. The coculture of LPS-treated Raw 264.7 cells with MSCs increased and decreased the expression of ARG-1 and M1 markers, respectively. Interestingly, the expression of ARG-1, an important anti-inflammatory cytokine, was reduced in diabetic kidneys, and it was increased by MSC treatment [39].

A previous study showed that treatment of *PTX-3*-silenced apoptotic macrophages with recombinant PTX-3 significantly reduced the secretion of inflammatory cytokines and activated the expression and secretion of anti-inflammatory cytokines [40]. Additionally, the therapeutic effects of MSCs were regulated by secreted PTX-3 in a rat bronchopulmonary dysplasia model [36]. The findings of previous studies and the present study suggest that PTX-3 is a primary factor involved in inhibiting inflammation and promoting an anti-inflammatory phenotype. In the present study, silencing of PTX-3 in SMUP-Cells, followed by their co-culture with macrophages, confirmed the role of PTX-3 secreted by SMUP-Cells in determining macrophage polarization, as evidenced by a significant decrease in the observed M2 phenotype (according to the levels of CD206 and ARG-1). PTX3 secreted by MSCs under an inflammatory condition activates the alveolar macrophage Dectin-1 receptor, suggesting that MSK1/2 controls macrophage activation l [36]. Dectin-1 phosphorylates the p38*α* MAPK or ERK1/2 cascade, leading to MSK activation [41]. Further, MSK1/2 induces the phosphorylation of transcription factor cAMP Response Element-Binding (CREB) to decrease inflammatory cytokine expression but controlling the secretion of anti-inflammatory cytokines [41,42].

Previous studies have used MIA-induced animal models to study arthritis pain [21,24]. MIA induces inflammation and pain by increasing the expression of nuclear factor-kappa B in synovial macrophages. Furthermore, another study indicated that anti-inflammatory drugs could inhibit degenerative changes and inflammation in joint tissues, as well as reduce pain [32]. In the present study, we established an OA model by administering high concentrations of MIA and confirmed that PTX-3 secreted by SMUP-Cell relieved the associated pain. This suggested that the anti-inflammatory effect of PTX-3 promoted the relief of OA-related pain. Reportedly, the Calcitonin gene-related peptide (CGRP) level has a positive correlation with pain [43]. Furthermore, substance P is involved in cold noxious pain and is associated with CGRP in inflammation-induced hyperalgesia [44], and tissue injury-induced pain; hence, it is a potential biomarker. 

The high MIA concentration used in this study is suitable for inducing pain, but there is a limitation in observing changes in joint structure owing to rapid cartilage erosion. In the future, we will evaluate the anti-inflammatory mechanism of action of SMUP-Cells and the mechanism of prevention of joint damage in a mild OA model established using low-concentration MIA. In vivo studies have shown that PTX-3 secreted by SMUP-Cells in an inflammatory environment represents a marker potentially capable of predicting the therapeutic efficacy of MSCs. Similarly, recent studies have demonstrated that the therapeutic effects of MSCs are indicated by the PTX-3 level in various in vivo models (e.g., wound healing, ischemic brain injury, and acute lung injury) [45,46,47]. These findings offer further evidence of the role of PTX-3 in the therapeutic efficacy of MSCs.

Various groups are conducting studies and clinical trials to develop a disease-modifying osteoarthritis drug (DMOAD) that can inhibit the progression of structural degenerative changes and improve the symptoms in OA [48,49]. Furthermore, animal study results of symptom relief and structural improvement of IA injection-type stem cell therapy have been reported. However, it is difficult to prove the efficacy of structural improvements in clinical trials. To overcome these limitations, it is necessary to develop an optimized product development strategy that strengthens the long-term survival function of cells and considers the therapeutic efficacy and cost between a single administration of high dose or repeated administrations of low doses [50]. In our study, we applied that HA should be pre-administered for the SMUP-Cells engraft. Currently, we are conducting clinical trials on patients with osteoarthritis Kellgren–Lawrence grades 2–3 using SMUP-Cells (NCT04037345, SMUP-IA-01). We plan to develop a successful therapeutic agent for osteoarthritis by proving the correlation between efficacy and mechanism of action by elucidating the effect of SMUP-Cells to improve cartilage structure through additional animal experiments and by obtaining clinical results from a large number of patients.

## 5. Conclusions

In summary, we described the development and application of SMUP-Cells as a next-generation stem cell-treatment technology platform via the improvement of stem cell life-cycle processes. The in vivo results showed that the administration of SMUP-Cells improved OA-related pain and that PTX-3 represents a marker for selecting cells with high potency. The findings of this study provide a new paradigm for the treatment of refractory inflammatory diseases by maximizing the effects of MSC therapy.

## Figures and Tables

**Figure 1 cells-10-02420-f001:**
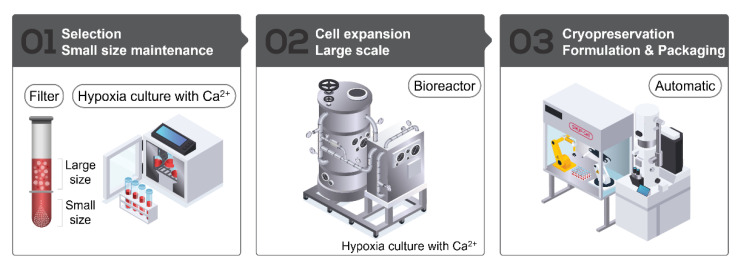
Summary of the SMUP-Cell platform. Selection step 01: isolation of small-sized cells cultured under hypoxic conditions in a calcium-rich medium. Cell-expansion step 02: scale-up using a bioreactor under the same culture conditions. Cryopreservation step 03: automated formulation and packaging with a xeno-free solution.

**Figure 2 cells-10-02420-f002:**
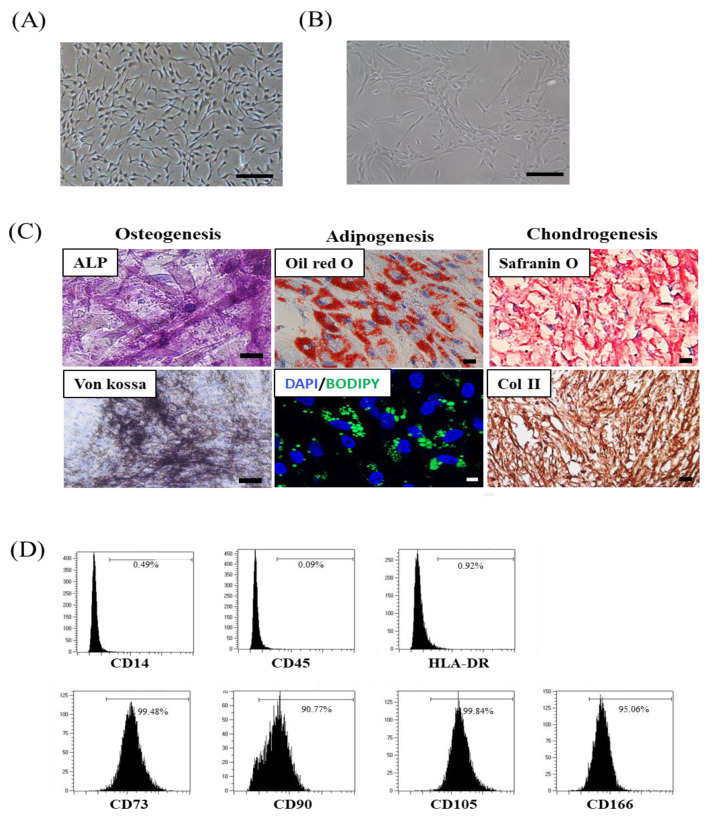
SMUP-Cell characterization. (**A**) Morphology shows adherent spindle-shaped cells at passage 6. (**B**) Cell staining measured SA β-gal as a senescence phenotype; it revealed no positive cells. (**C**) During incubation in a specialized induction medium, multi-lineage potential was demonstrated by staining typical multi-lineage markers. Osteogenic cells were analyzed according to ALP and von kossa levels. Adipogenic cells showed increased lipid vacuoles within the cytoplasm via Oil Red O and BODIPY (green) staining. Chondrogenic cells accumulated sulfated proteoglycans stained with Safranin O. Nuclei were counterstained with hematoxylin. Nuclei were stained with DAPI (blue). The merged image is an overlay of the DAPI and BIODIPY images. Scale bar = 100 µm. (**D**) Flow cytometric analysis of cells based on the cell-surface expression of typical MSC markers. Cells were strongly positive for the MSC markers CD73, CD90, CD105, and CD166 and negative for CD14, CD45, and HLA-DR.

**Figure 3 cells-10-02420-f003:**
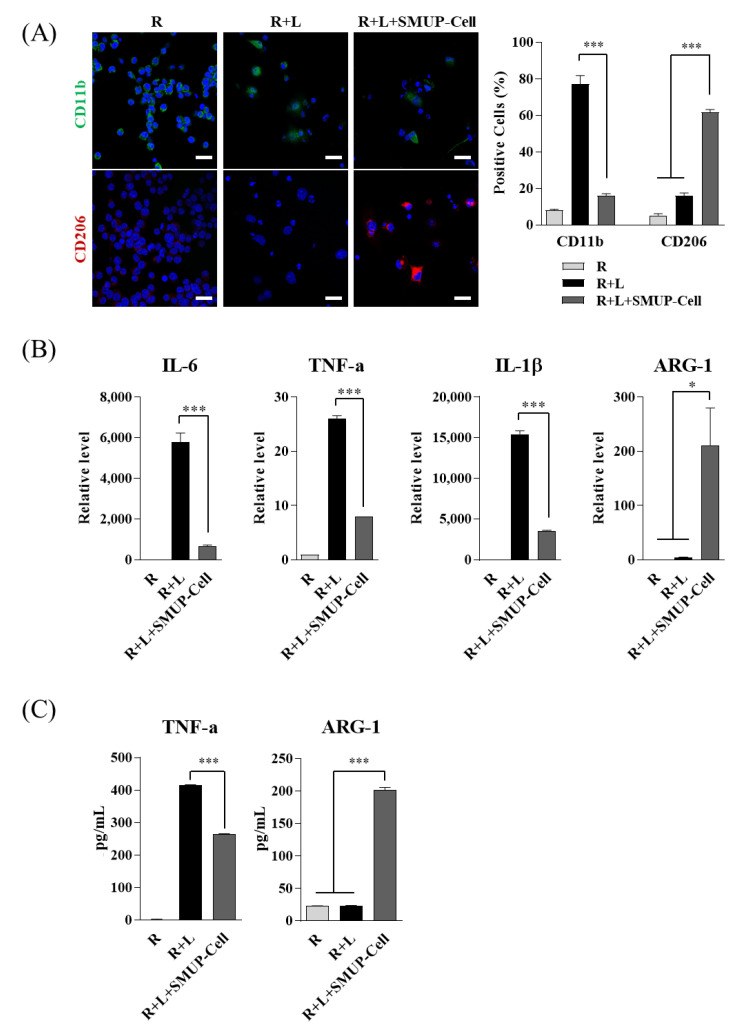
Macrophage polarization by SMUP-Cells under LPS-stimulated inflammation conditions. Mouse macrophages (Raw 264.7) were exposed to LPS and co-cultured with SMUP-Cells for 2 days. (**A**) The M1 and M2 phenotypes were evaluated by staining CD11b (green) and CD206 (red), respectively, followed by confocal microscopy. Nuclei were stained with Hoechst 33342 (blue). Quantitative data represent the percentage of CD11b^+^ or CD206^+^ cells. Scale bar = 100 µm. Confocal microscopy of co-cultured macrophages showed a significant decrease in the M1 marker CD11b and an increase in the M2 marker CD206. (**B**) Quantification of gene expression levels of inflammatory cytokines (mouse IL-1b, IL-6, and TNF-α) and anti-inflammatory cytokines (mouse ARG-1) was determined by qRT-PCR. Levels were normalized to those of β-actin, with expression levels in naïve cells set at 1. (**C**) Secretion of TNF-α and ARG-1 was evaluated by ELISA. Medium from SMUP-Cell co-cultures showed significantly decreased levels of TNF-α and elevated levels of ARG-1. (**A**–**C**) Data represent mean ± SD (n = 3 per group). * *p* < 0.05, *** *p* < 0.001. R, Raw 264.7 cells; L, LPS treatment.

**Figure 4 cells-10-02420-f004:**
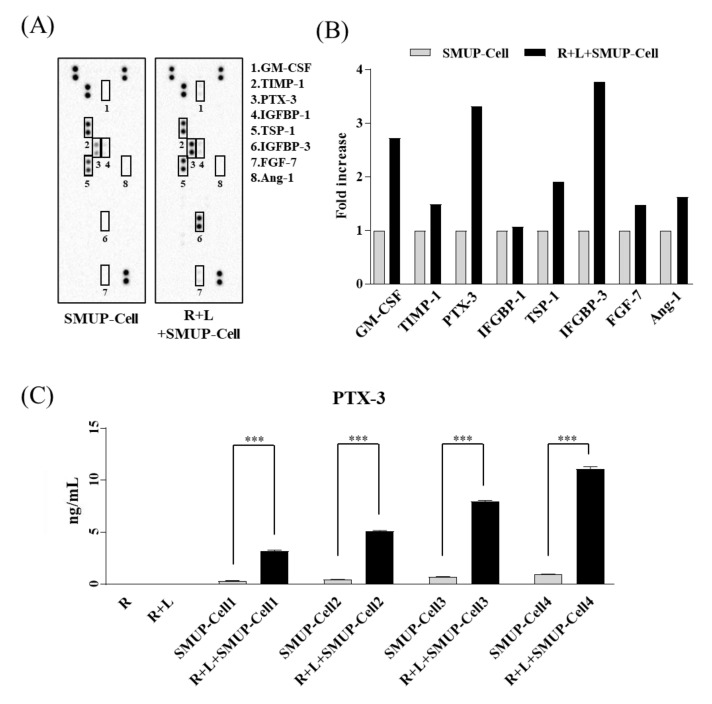
Array to evaluate the changes in SMUP-Cell protein secretion under inflammatory conditions. (**A**) Array analysis using supernatants from SMUP-Cell cultured alone versus SMUP-Cells co-cultured with LPS-exposed Raw 264.7 cells. Secreted proteins were characterized using a protein array (55-spot, n = 1). Eight proteins in the membrane were upregulated in the co-culture group (squares). (**B**) Quantification of optical intensity of anti-inflammatory factors showed upregulated protein secretion in the co-culture relative to that in the culture of SMUP-Cells. Protein levels were evaluated according to fold increase, with data normalized to levels from a culture of SMUP-Cells alone (set to 1). (**C**) To confirm the upregulation of PTX-3, its level was analyzed by ELISA in supernatants from cultures of SMUP-Cells alone and co-cultures in four different lots. Data represent mean ± SD (n = 3 per group). *** *p* < 0.01. R, Raw 264.7 cells; L, LPS treatment.

**Figure 5 cells-10-02420-f005:**
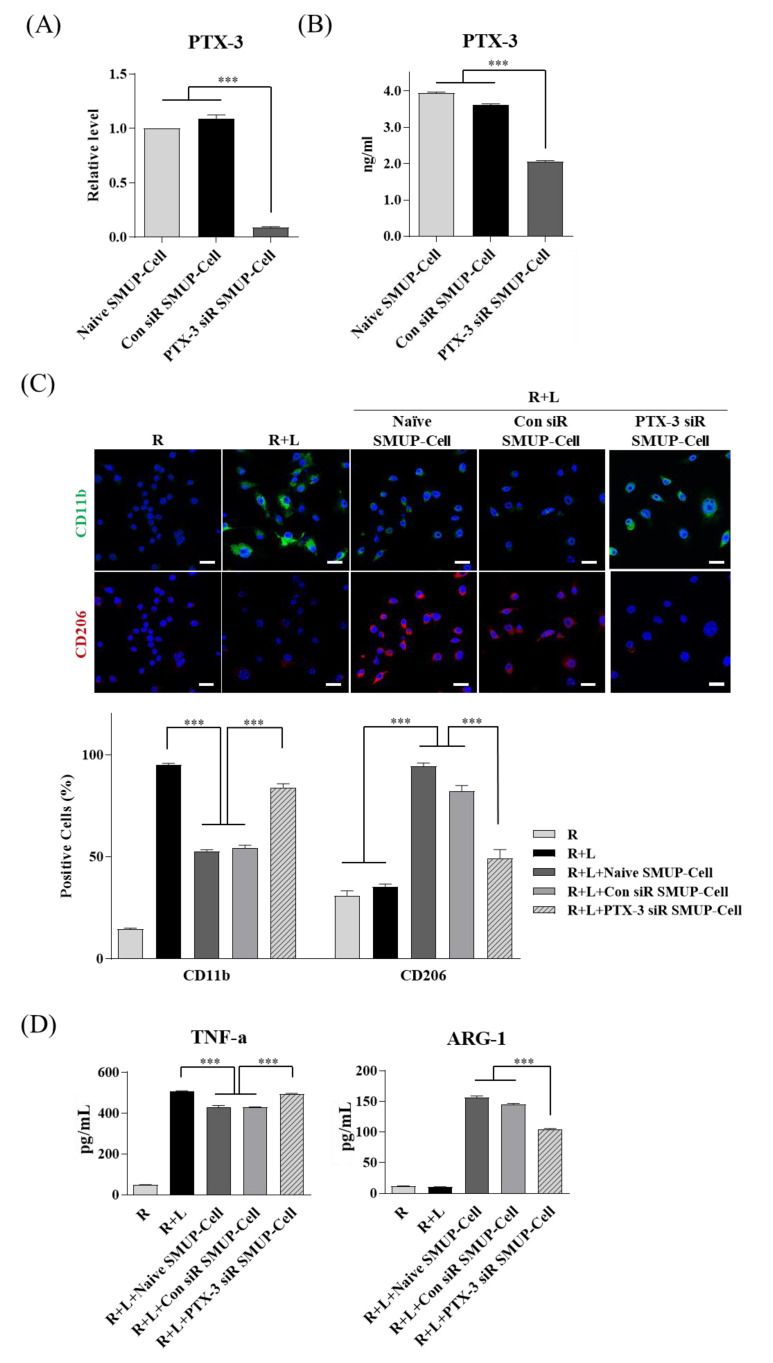
PTX-3 silencing in SMUP-Cells accelerates macrophage polarization. Raw 264.7 cells were exposed to LPS and then co-cultured with SMUP-Cells for 2 days. SMUP-Cells were transfected with control siRNA or PTX-3 siRNA before co-culture. (**A**) Gene expression and (**B**) secreted protein levels of PTX-3 were compared relative to those in non-transfected SMUP-Cells. (**C**) Expression of CD11b (green) and CD206 (red) was evaluated by quantifying the percentage of positively stained cells. Nuclei were stained with Hoechst 33342. Scale bar = 100 µm. PTX-3 siRNA-transfected cells demonstrated altered macrophage polarization phenotype. Confocal microscopy of LPS-stimulated Raw 264.7 cells co-cultured with PTX-3-silenced SMUP-Cells revealed significantly upregulated expression of CD11b and attenuated expression of CD206. (**D**) ELISA demonstrated lower levels of TNF-α, inflammatory cytokine, and higher levels of ARG-1, anti-inflammatory cytokine, in medium from PTX-3-silenced SMUP-Cells. Error bars represent mean ± SD (n = 5 per group). *** *p* < 0.001. R, Raw 264.7 cells; L, LPS-treatment.

**Figure 6 cells-10-02420-f006:**
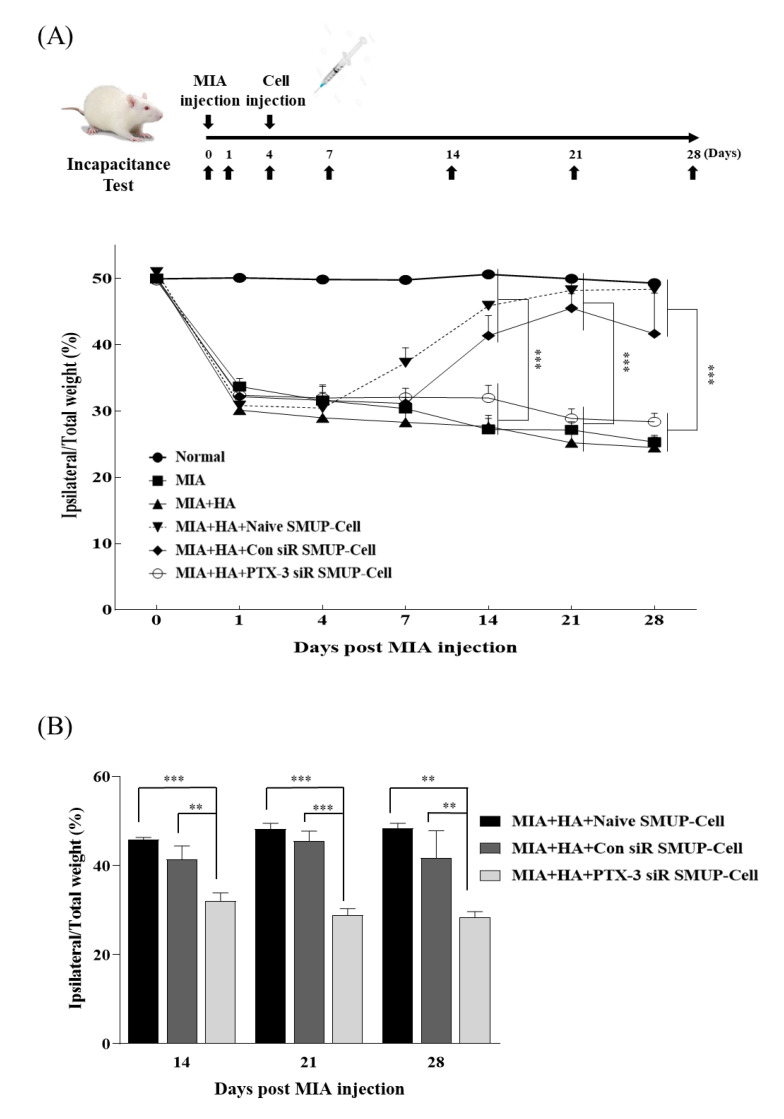
Effect of PTX-3 secreted by SMUP-Cells on pain in the MIA-induced OA rat model. The efficiency of siRNA transfection of SMUP-Cells with scrambled-siRNA (Con siRNA) or PTX-3 siRNA for 24 h. (**A**) Changes in body weight distribution between the hind limbs were calculated as ((weight borne on ipsilateral paw / sum of the weight borne on the ipsilateral and contralateral paws) × 100) and evaluated on days 0, 1, 4, 7, 14, 21, and 28 after the injection of 2 mg MIA. In all groups, HA and the respective cells were IA administered into OA rat knees at 4 days after MIA injection. (**B**) Analysis of body weight distribution in the SMUP-Cell treated OA rat knee at 14, 21, and 28d after MIA injection. Error bars represent mean ± SEM (N = 6 per group). ** *p* < 0.01, *** *p* < 0.001. HA, hyaluronic acid; MIA, monosodium iodoacetate.

## Data Availability

The data presented in this study are available on request from the corresponding author.

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
