# Peer review of "PTX-3 Secreted by Intra-Articular-Injected SMUP-Cells Reduces Pain in an Osteoarthritis Rat Model"

_cells, 2021, doi:10.3390/cells10092420_

Round 1
Reviewer 1 Report
The manuscript by Lee et al. titled "PTX-3 secreted by intra-articular-injected SMUP-cells reduces pain in an osteoarthritis rat model" is an interesting work using MSCs to reduce pain. The work is relatively well written and results are interesting to readers. However, the manuscript would benefit from addressing the following points:
1) The Abstract is contradicting itself. In one sentence high PTX-3 is important in reducing the pain whereas in another sentence silencing PTX-3 reduces pain. Please fix according to the results.
2) Why is the name SMUP-cell needed? These are just hypoxia activated MSCs produced en masse.
3) The text size is not consistent in the figures.
4) Some figures the text is too small to read.
5) Why is the siRNA efficacy data in Figure 6 and not in Figure 5 as siRNA is first introduced in Figure 5?
6) Based on Figure 5b, PTX-3 is not a key factor in the process. It does contribute to it but most of the effect is by other molecules.
7) In Figure 6, it would be better to show the whole y-axis scale and then an Zoomed-in inset as shown so the reader can understand the scale better.
Author Response
We thank you for your thorough review. Your comments have helped us improve the quality of the manuscript. We have revised the manuscript according to your comments. Our responses to the comments are provided attatched file.

Reviewer 2 Report
In this work, Lee and col. show that a subpopulation of MSCs, the SMUP-cells, can polarize the macrophage phenotype towards the anti-infammatory M2 phenotype and that this process is mediated by an increased secretion of PTX3. They demonstrate that intra-articular injection of those cells have a beneficial effect on pain managemen of OA.
The authors describe the selection of the SMUP cells and the implementation of a platform for cell processing that would improve standardization of production.
This is an interesting work that could have important implications in the clinic. However, there are major issues that would preclude its publishing in its current form.
Major points.
The sentence in the abstract starting “To identify the paracrine action…..”should be divided in two for clarity. The part referring to the mRNA silencing validation should be an independent sentence.
Line 33: Please, change “target cells” for “cell lineages”
Lines 38 to 42: please, specify in which cell types or microenviroment were those effects reported.
In line 44 the authors state that MSCs have been proven useful to regenerate cartilage in OA. As far as this reviewer is aware of, MSC seem to delay degeneration more than activate regeneration. If the authors know of studies where true cartilage regeneration has been demonstrated after MSCs injection, they should provide adequate references.
The protocols at sections 2.1, 2.2, 2.3, 2.4, 2.5 and 2.7 are not detailed enough. In brief:
2.1. An estimated of the number of cells plated, recovered, and isolated after filtering should be given. Also, a flow cytometry profile, showing the characteristics (surface markers) of the isolated subpopulation must be included.
The authors should indicate whether all the cells from the five donors were pulled together.
2.2. Working dilutions of the different antibodies used should be given. Why did the authors fix the cells? Was that done prior to flow cytometry analysis? Was not possible to do this with live cells? Composition of the induction media and extent of the induction should be clearly specified.
2.3.incubation time with lipopolysaccharide should be given. Is this added directly to the culture media? Is LPS kept in the media during the co-culture? Which media is used to collect the supernatants?
2.4. Antibody dilution and solution used for hybridization and blocking solutions should be given. A more detailed protocol is needed.
2.5 How much secretome was used for this assay? Was the secretome concentrated? How long was the media incubated with the cells.
2.7. Total number of animals used in the experiment should be clearly defined. Which volume of the MIA solution was injected? Did the treatment consist of one unique dose of MIA? Cells were used for injection right after they were thawed with no previous culture? Why? Why weren’t the cells administered together with the hyaluronate? What is the rational to do two separate injections in this case? What was the volume of hyaluronate injected?
Traditionally, MSCs undergo replicative senescence after only a few passages. It is therefore key to check for senescence markers in those cells prior to use. While the authors did performe a b-galactosidase assay, this clearly lacks appropriate (positive) controls. Besides, additional analysis should be performed, such as measuring p16 INK4A expression or immunohistochemistry of the senescence associated heterochromatin foci, also using appropriate controls.
Information concerning the results of the differentiation analysis are less than optimal. Images provided lack enough quality. As an example, adipogenic staining with oil red should be showing a much bigger area to show that a high percentage of the cells in culture have this potential. It is difficult to believe that the same magnification was used for all the pictures. Is this correct? Proving the maintenance of multilineage differentiation potential is key when working with MSCs. Additional assays such as measuring the main osteogenic, adipogenic and chondrogenic markers should be performed. Without clear data it is hard to believe that the SMUP-cells have stem cells characteristics.
Although it is true that the authors have described the surface markers characteristics of the so called “small-sized cells” in a previous publication, it is key that they show a flow cytometry profile of all the MSC population with the relevant markers, specifically pointing out which sub-population was used for the study. This is important to know what we are dealing with. Also, for all the flow cytometry analysis proper positive and negative controls should be included, although this can be done in supplementary information.
This review finds surprising that the estimation of cells positive for CD11b or CD206 in figures 3 and 5 was done by scoring the positive cells by confocal microscopy rather than performing a flow cytometry analysis using those same antibodies. Both, CD11b and CD206 are markers frequently used in flow cytometry. Could the authors explain this?
Line 262. The authors should specify if the Raw cells had been previously treated with LPS.
How many times was the array performed? Was this a one-time experiment? This is not specified in the figure legend. Since the anti-inflammatory effect is likely to be mediated by the increase of more than one factor in coculture, it is important that the authors evaluate the effect of the other two anti-inflammatory factors since all of them seem to be upregulated to some extent? If the authors did not perform those measures at least they should explain the rationale of disregarding the results obtained with the other relevant anti-inflammatory factors. Only a brief mention about these other factors is done in lines
Lines 294 to 295. The authors mention in the text that they analyzed “levels of morphology”. What are they referring to?
Line 301. The authors relate PTX3 silencing to an increase in inflammatory cytokines and a decrease in anti-inflammatory cytokines. Could the authors offer an explanation to this? Are there previous data in the literature regarding a crosstalk between those factors? The authors should at least discuss this.
Line 327. The authors indicate that cells were stained with Neostatin prior to their injection in the knee. However, this procedure is not described in materials and methods. A detailed description of the staining procedure and materials used should be provided in that section.
The authors refer to a rat model of “severe OA”. What are the characteristics of this model? Does is refer to the time passed since the MIA injection? How do the authors evaluate the severity of the model?
The authors should specify the meaning of “naïve” cells the first time this is used in the text.
Line 372. The simple culture methods used for MSC is not limiting their use in the clinic. Quite the opposite. Please rephrase.
Lines 437 to 439. The authors do not provide any proof of this statement. They did not perform a histological study on the structure of the knee cartilage after the treatment. This is a key part of the study that has not been done. Since inflammation is known to be a key part of the degeneration process, the injection of the cells should have a positive effect or at least delay this degeneration, but no studies were done in this respect. A histological analysis of the knee before and after the treatment must be included to be able to claim that their cells inhibit OA progression.
The intra-articular injection if MSCs has several drawbacks as widely addressed in the literature. These should be at least briefly addressed in the discussion.
There are several grammatical and spelling mistakes throughout the manuscript that need to be corrected.
Author Response
We thank you for your thorough review. Your comments have helped us improve the quality of the manuscript. We have revised the manuscript according to your comments. Our responses to the comments are provided attached file.

Reviewer 3 Report
The authors of the study have investigated the application of their SMUP cells in OA model with an emphasis on the immunomodulatory phenotype initiated by these cells. Co-culture with macrophages resulted in a reduction in inflammatory cytokines and increase in anti-inflammatory molecules with macrophages converting to their M2 phenotype. One particular molecule, PTX-3, was found to be critical in the process. Use of SMUP cells containing siRNA of PTX-3, showed that secretion of this molecule significantly increased pain in an OA induced mouse model, demonstrating the importance of this molecule.
The authors show the importance of PTX-3 for treating OA pain. The authors need to answer the following,
1) At what oxygen tension was the differentiation described in figure 2c undertaken ?
2) Figure 6c needs to be presented in a different way, as it is difficult to see differences between groups.
3) Authors should present the sections of treated mice from each group to show if there was improvements in cartilage healing ? If there are enough sections, OARSI score for cartilage regeneration should be performed.
4) In this case, pain has been measured by the authors. They should also measure pain receptors or secreted pain molecules (e.g. alpha CGRP) to further clarify their in vivo results on a molecular level.
Author Response
We thank you for your thorough review. Your comments have helped us improve the quality of the manuscript. We have revised the manuscript according to your comments. Our responses to the comments are provided file.

Round 2
Reviewer 1 Report
Thank you for improving the manuscript based on my comments. The following points should still be addressed though:
- This sentence in the abstract is still unclear: "PTX-3 silencing in SMUP Cells significantly decreased their therapeutic effects against monosodium iodoacetate (MIA)-induced OA, resulting in reduced pain relative that in the control group." The current version of the sentence is saying that silencing PTX-3 reduced pain. I do not think that is what it should say.
- I still question the need for giving these cells a novel name confusing the MSC field even further, but I leave that up to the Editors for consideration.
Author Response
The responses to all the comments have been prepared and attached herewith.

Reviewer 2 Report
This reviewer has no further questions
Author Response
We thank you and the reviewers for the valuable comments and suggestions.
Reviewer 3 Report
The authors have responded to my questions appropriately. May I ask that they state the details described in question 1 and include it in figure 1.
Author Response
Thank you again for the thorough review of our manuscript and we really appreciate all the suggestions. We added differentiation potential on Fig. S1 (normoxia vs. 3% hypoxia condition), as followed by the reviewer’s suggestion to improve the quality of the manuscript.